# Multi-omics delineation of cytokine-induced endothelial inflammatory states

Stijn A. Groten[1], Eva R. Smit[1], Esmée F. J. Janssen[1], Bart L. van den Eshof [1], Floris P. J. van Alphen[1], Carmen van der Zwaan[1], Alexander B. Meijer[1,2], Arie J. Hoogendijk [1,3] & Maartje van den Biggelaar [1,3✉]

Vascular endothelial cells (ECs) form a dynamic interface between blood and tissue and play a crucial role in the progression of vascular inflammation. Here, we aim to dissect the system-wide molecular mechanisms of inflammatory endothelial-cytokine responses. Applying an unbiased cytokine library, we determined that TNFα and IFNγ induced the largest EC response resulting in distinct proteomic inflammatory signatures. Notably, combined TNFα + IFNγ stimulation induced an additional synergetic inflammatory signature. We employed a multi-omics approach to dissect these inflammatory states, combining (phospho-) proteome, transcriptome and secretome and found, depending on the stimulus, a wide-array of altered immune-modulating processes, including complement proteins, MHC complexes and distinct secretory cytokines. Synergy resulted in cooperative activation of transcript induction. This resource describes the intricate molecular mechanisms that are at the basis of endothelial inflammation and supports the adaptive immunomodulatory role of the endothelium in host defense and vascular inflammation.

[1] Department of Molecular Hematology, Sanquin Research, Amsterdam 1066 CX, The Netherlands. [2] Department of Biomolecular Mass Spectrometry and Proteomics, Utrecht Institute for Pharmaceutical Sciences (UIPS), Utrecht University, Utrecht 3584 CS, The Netherlands. [3] These authors contributed equally: Arie J. Hoogendijk, Maartje van den Biggelaar. ✉email: m.vandenbiggelaar@sanquin.nl

Endothelial cells (ECs) line the inside of our blood vessels and form a dynamic interface between blood and surrounding tissues. Apart from facilitating oxygen, nutrient, and waste product exchange, ECs control hemostasis by attracting platelets to seal breaches in the vascular walls during primary hemostasis[1]. Moreover, ECs are crucial gatekeepers controlling the trafficking of immune cells into and out of tissues during inflammation. For their role in this adaptive synapse, ECs are well-equipped to sense environmental cues, such as mechanical stress, hormones (e.g., vasopressin, histamine), cells (e.g., neutrophils, monocytes, platelets) and other external stimuli (e.g., thrombin, cytokines)[2–5]. In addition to the transmigration of immune cells, ECs have several immunomodulatory capacities such as antigen presentation and cytokine secretion[5,6]. However, although ECs carry these immune-modulating properties and are among the first cells to come into contact with pathogens, they are rarely mentioned in immune cell networks[7–9].

Deregulation of EC homeostasis can result in over-inflammatory or hyper-coagulation states of the endothelium. This endothelial dysfunction is implicated in several multi-facetted inflammatory diseases, including transfusion related acute lung injury, sepsis, rheumatoid arthritis, acute respiratory distress syndrome (ARDS), eye vasculopathies, chronic kidney disease and COVID-19[10–19].

Although both endothelial homeostasis and cytokines are deregulated in these diseases, the molecular basis which orchestrates adaptive endothelial-cytokine interactions is mostly confined to research on tumor necrosis factor-alpha (TNFα). Moreover, synergism between cytokines such as TNFα and interferon-gamma (IFNγ) has been observed in ECs and linked to detrimental effects in inflammatory disorders[20–23]. Although underlying mechanisms have been proposed, a system-wide EC response has not been characterized.

Therefore, in this study, we set out to dissect the molecular signatures of endothelial-cytokine responses, employing blood outgrowth endothelial cells (BOECs), also known as endothelial colony forming cells, as our source of ECs because of their extensive, robust expansion, expression of mature vascular EC markers and ability to be isolated from adult donors[24,25].

We show that ECs express the receptor-repertoire to facilitate various cytokine signals. However, upon stimulation with an unbiased cytokine library, we observed predominantly unique inflammatory states for TNFα and IFNγ. Moreover, combined stimulation of TNFα and IFNγ resulted in a synergetic EC response. Combining multiple omics levels, we dissected the molecular basis of these inflammatory states from signaling (phosphoproteome) to mRNA transcription (transcriptome), protein regulation (proteome) and protein secretion (secretome). This study reveals system-wide adaptive EC inflammatory states, emphasizing the role of EC-cytokine interactions in inflammatory pathogenesis and reiterating ECs as an adaptive player in inflammation.

## Results

**Mapping cytokine-endothelial interactions**. Although vascular inflammation is orchestrated by cytokines that activate the endothelium, knowledge on endothelial-cytokine interactions at the system level is limited. To review established interactions, we used immuneXpresso, a text mining engine that extracts directional cell-cytokine interactions from PubMed abstracts[26]. Of the 143 cytokines present in this database, 65 were identified to interact with ECs. Most studies report the interaction with cytokine TNFα and fibroblast growth factor 2 (*FGF2*) (Fig. 1a). To determine whether this is due to a confirmation bias in literature or whether ECs do not express the receptors to interact with other cytokines, we performed an in-depth proteomic analysis of BOECs to determine the receptor-repertoire. Out of 6848 quantified proteins, we found 166 receptors with known ligand interactions[8] (Fig. 1b and Supplementary Fig. 1a), showing that ECs possess a wide array of receptors that can interact with ligands present in their microenvironment.

**Proteome response profiling of EC-cytokine interactions**. Next, we profiled EC responses to different cytokines and stimulated BOECs with a library of 92 signaling proteins based on commercial availability. This library contained 46 proteins from the ImmuneXpresso search including both TNFα and *FGF2* (Supplementary Table 1). BOECs were stimulated for 24 h and proteomics responses were analyzed using high-resolution label-free quantitative (LFQ) MS. We quantified a total of 4790 proteins with an overall median Pearson correlation coefficient of 0.96 (Supplementary Fig. 2). Of the 92 signaling proteins, 47 had an impact on the proteome and in total 293 proteins were differentially abundant (Fig. 2a). TNFα stimulation induced the strongest response, followed by IFNγ, IL1β and IL1α. To visualize response similarities between stimuli, we constructed a network of the number of differentially abundant proteins and overlap per stimulus (Fig. 2b and Supplementary Fig. 3a). This revealed TNFα as a central "knot" overlapping with 7 other cytokines. The highest similarity was observed between TNFα, IL1α and IL1β. We also observed an

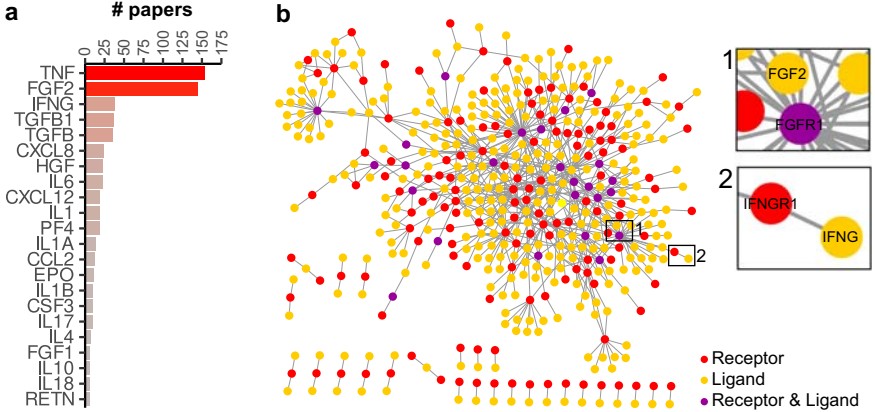

**Fig. 1 Knowledge-based mapping of cytokine-endothelial interactions. a** Bar plot based on immuneXpresso data mining showing the number of papers describing interactions between the cytokines and endothelial cells (# papers > 5). **b** Cytoscape interaction network of receptors and potential ligands (red dots: receptors, yellow dots: ligands, purple dots: proteins fulfilling both receptor and ligand criteria), edges represent STRING-DB scores. Inserts show zooms of example cytokine-receptor interactions; for network with labels, see Supplementary Fig. 1b.

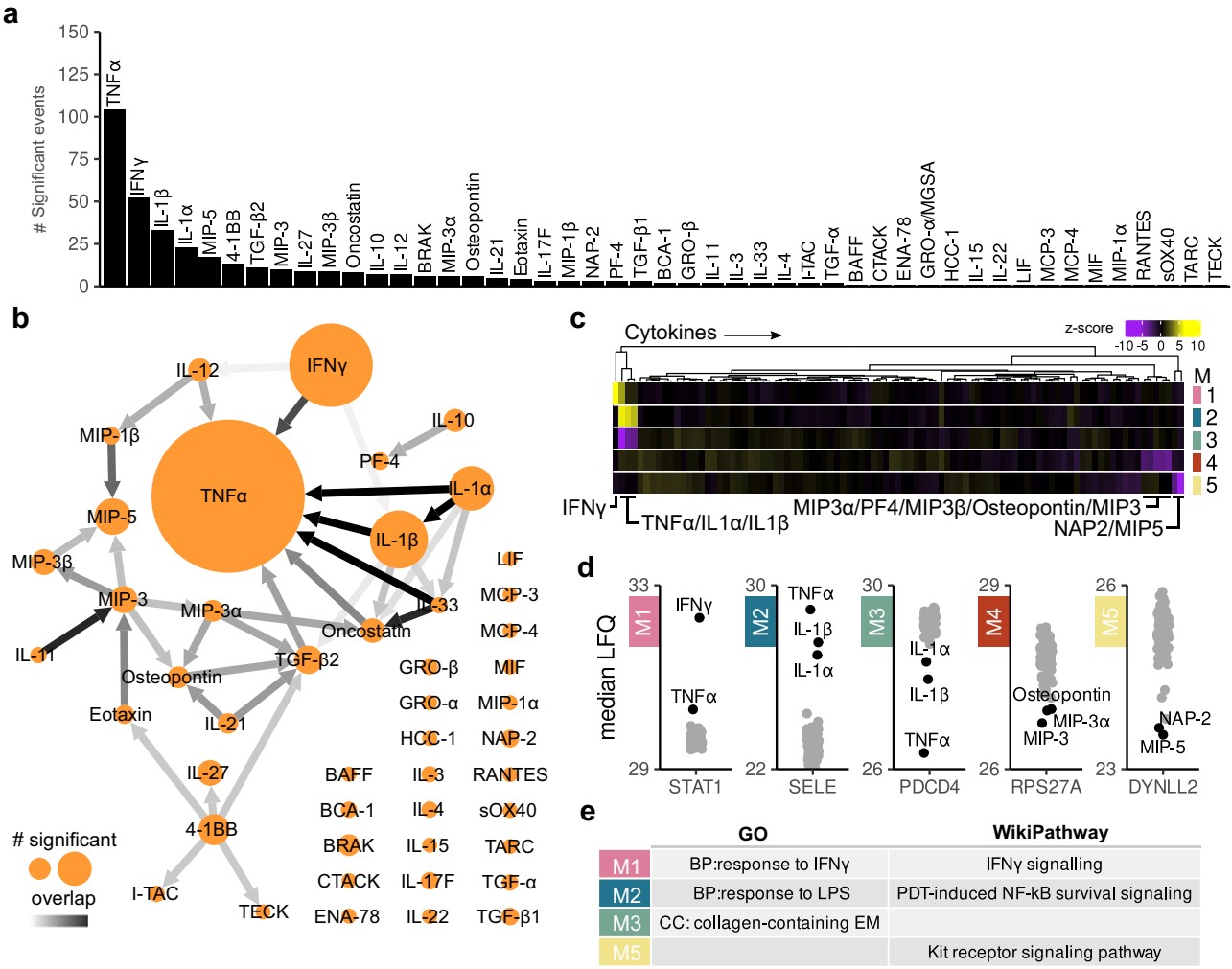

**Fig. 2 Proteome response profiling of EC-cytokine interactions. a** Bar plot showing the amount of differentially regulated proteins for the stimuli that altered the BOEC proteome after 24 h of stimulation (moderated *t*-test, Benjamini–Hochberg (BH) adjusted *p* value < 0.05 and log2 fold change > 1). **b** Summarizing network of differentially abundant proteins between stimuli. Node labels show cytokine stimuli. Node size represents amount of statistically significant proteins. Edges show overlap between proteomes, color intensities (white to black) of edges indicate amount of overlapping proteins as a ratio of the smaller node. See Supplementary Fig. 3 for non-summarized network. **c** Profile plots of modules describing cytokine proteomic responses with cytokine annotation. Gradient scale indicated *z*-scores of median LFQ-score of genes in a module per stimulus, Yellow: cytokines related to an increased abundance response profile; purple: cytokine(s) related to a decreased protein abundance response, cytokines which contribute to the module regulation are highlighted. Replicates have been summarized to medians for visualization, modules are indicated by color, M1 (pink), M2 (blue), M3 (green), M4 (red) and M5 (yellow). **d** Proteins with high modules membership scores plotted as median label-free intensities (LFQ). **e** Enriched GO terms and Wikipathways per module. MF molecular function, CC cellular component, BP biological process.

overlap between IL-33-TNFα, IL-33-Oncostatin and IL-11-MIP3. Co-expression-based clustering delineated 5 dominant EC response types (Fig. 2c). These consisted of two modules with increased proteins abundances (Modules 1 and 2: IFNγ and TNFα/IL1α/IL1β responses) and three reduced protein abundance modules (Modules 3-5, representing TNFα/IL1α/IL1β, MIP5/NAP2 and MIP3/MIP3B/MIP3A/Osteopontin/PF4). Proteins with high module memberships, highlighted hallmark IFNγ and TNFα responsive proteins Signal transducer and activator of transcription 1-alpha/beta (*STAT1*) and E-selectin (*SELE*) for modules 1 and 2 (Fig. 2d). Gene Ontology (GO) enrichment and pathway analysis[27] revealed that module 1 enriched for "response to IFNγ" and "IFNγ signaling" as expected, while module 2 consisted of "response to LPS" and NF-κB signaling. Module 3 enriched for extracellular component "collagen-containing extracellular matrix" and module 5 for "Kit receptor signaling pathway". There was no significant enrichment for module 4 (Fig. 2e and Supplementary Fig. 3b, c).

**Combined TNFα and IFNγ stimulation induces synergetic EC effects.** As TNFα and IFNγ induced the highest, predominantly upregulated, responses and synergism between TNFα and IFNγ has been reported, we assessed the effects of combining both cytokines. Initially, we studied EC morphology, and as expected, TNFα-stimulated cells shifted from a cobblestone round-like morphology to an elongated shape (Fig. 3a). Although IFNγ did not induce observable changes, the combination of both stimuli resulted in an amplified TNFα morphology, in which all ECs changed to an elongated shape and a more contracted monolayer. Guided by these observations, we tested whether the proteome was impacted similarly using LFQ MS. Initially, we performed a concentration range of TNFα and IFNγ separately and observed similar proteomic profiles for stimulation at 10 and 100 ng/ml (Supplementary Fig. 4). Stimulation with TNFα, IFNγ and in combination resulted in three distinct signatures (Fig. 3b). Compared between each other, TNFα and IFNγ induced 74

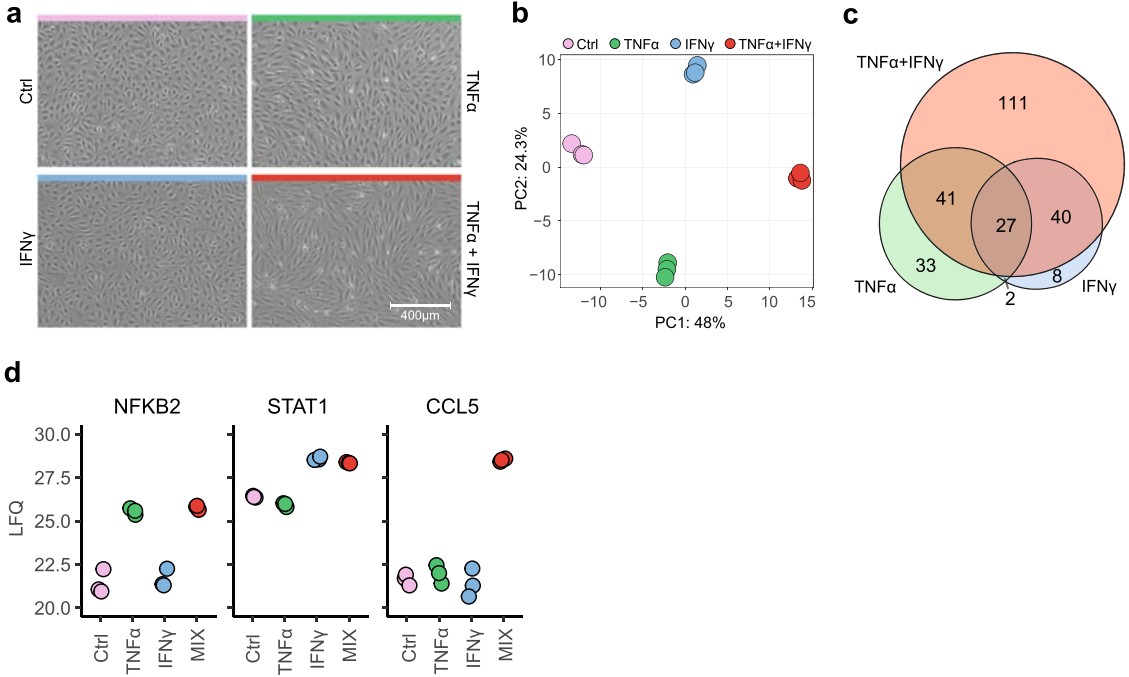

**Fig. 3 Synergistic proteomics response of TNFα and IFNγ in ECs. a** Brightfield images of unstimulated ECs (Ctrl, pink) or ECs stimulated with TNFα (green), IFNγ (blue) or TNFα + IFNγ (red) for 24 h. **b** Principal component analysis (PCA) of proteomes of ECs stimulated with TNFα, IFNγ and TNFα + IFNγ. **c** Euler plot of total amount of unique and overlapping differentially regulated proteins per stimulus (moderated *t*-test, BH-adjusted $p < 0.05$ and log2 fold change > 1). **d** LFQ intensities of hallmark proteins per stimulation.

versus 48 unique proteins, respectively, while combined stimulation induced 111 unique altered protein abundances (Fig. 3c). Unique proteins per stimulus included transcription factors *NFKB2* and *STAT1* for TNFα and IFNγ respectively and chemokine *CCL5* for combined stimulation (Fig. 3d).

**Temporal dynamics of TNFα and IFNγ stimulation on phosphoproteome, transcriptome and proteome reveal systemwide inflammation states.** Prompted by the distinct proteomic signatures of TNFα, IFNγ and combined stimulation and the notion that these cytokines are an object of extensive study in ECs, we aimed to dissect the molecular basis of the observed inflammatory states. To this end we employed several omics levels (phosphoproteome, transcriptome and proteome) in a time-resolved experiment to delineate the cytokine responses (Fig. 4a). To accurately quantify phosphopeptide levels we utilized a SILAC-MS workflow. Phosphoproteomes and proteomes were acquired from the same sample, whereas transcriptomic data were obtained from parallel stimulations. We quantified 4171 proteins, 6144 phosphosites and 60,664 transcripts. On all levels, TNFα + IFNγ stimulation increased the number of significant events, compared to single stimuli (Fig. 4b). Transcript levels had the largest increase in events after combined stimulation compared to single stimuli (3.7-fold), while proteomic events increased less (2.3-fold). To visualize the regulatory dynamics between different omics levels, we evaluated the timing of these events. In line with the type 1–2 EC activation paradigm[28], phosphoregulation occurred first (0–30 m), followed by transcriptomic (4–12 h) and proteomic events (8–24 h) (Fig. 4c). Furthermore, this analysis showed different dynamics between stimuli. The number of TNFα-induced transcriptomic events decreased after 12 h, while for IFNγ, this number remained stable. Next, we performed GO-term enrichment analyses and compared enriched terms between omics levels and stimulations. Early responses, characterized by changes in phosphosites, reflected

mostly mechanical changes (e.g., cadherin binding and focal adhesion) and were shared between all stimulations (Fig. 4d). Enrichment of differentially regulated mRNAs included GO-terms "ECM organization" and "growth factor binding", while proteome enriched for antigen presentation processes such as "MHC protein complex", "peptide antigen binding" and the generic term "response to IFNγ". To show the temporal nature of the omics levels, we plotted examples of each: phosphosite *ARHGEF10* S379, and the transcript and protein levels of *ICAM1* (Fig. 4e). The *ARHGEF10* phosphosite was dephosphorylated within 2 min, while mRNA levels of *ICAM1* peaked from 30 min to 4 h and approached baseline after 24 h. This pattern was followed by a steady increase in protein level reaching its peak at 24 h. To assess how the steady-state repertoire of BOECs compared to other EC types, we compared our RNAseq data to three published studies on various cultured primary ECs within the endoDB[29–35]. Principal component analysis showed high overlap with HUVECs in Rombouts et al.[32,34] and BOECs and pulmonary ECs in Long et al.[31,35] (Supplementary Fig. 5a). However, the correlation of transcriptome signatures also highlighted study-induced variation (Supplementary Fig. 5b). Relative expression levels of key EC genes as well as TNFα and IFNγ receptors were similar between the majority of EC types (Supplementary Fig. 5c).

**Visualizing endothelial inflammatory states.** To visualize the processes that are driven by either TNFα, IFNγ or combined stimulation, we generated a map of EC responses. First, we classified all differentially regulated phosphosites, transcripts and proteins as "TNFα", "IFNγ", "common", "TNFα + IFNγ" or "not classified", based on effect sizes and dynamic behavior in time (Fig. 5a). For phosphosites most hits were classified as "common", while for both mRNA and protein "IFNγ" was the most prevalent classifier (1850 transcripts and 60 proteins), followed by "TNFα + IFNγ" (1235 and 45) and "TNFα" (948 and 34) (Fig. 5b). As an example of each classification, we plotted highly

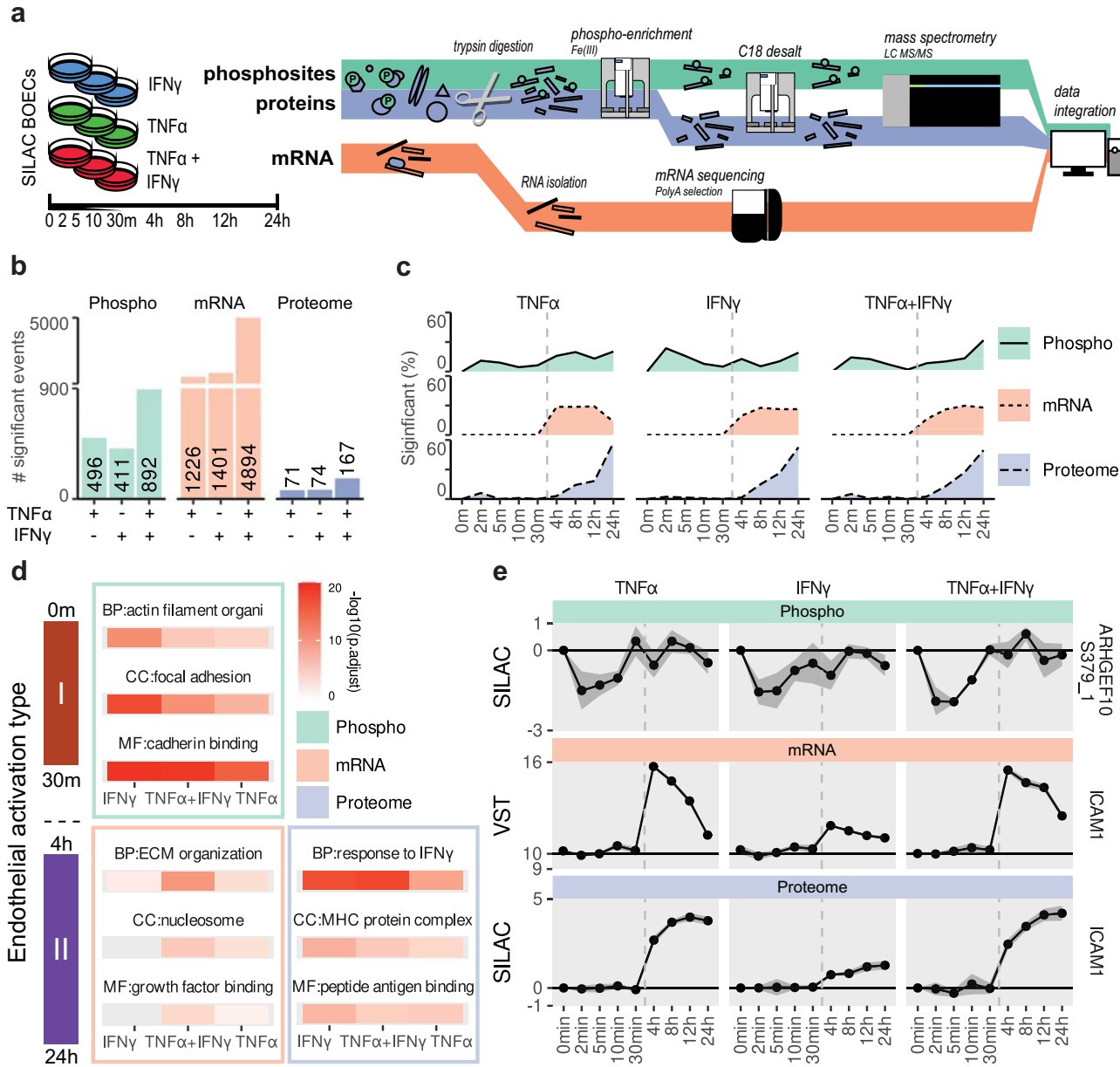

**Fig. 4 Temporal multi-omics analysis of TNFα, IFNγ and combined stimulation responses. a** Schematic overview of cell stimulation and multi-omic workflow. **b** Bar plot of total statistically significant events per stimulus on phosphoproteome (teal), transcriptome (orange) and proteome (purple) levels (moderated *t*-test, BH-adjusted *p* < 0.05 and log2 fold change > 1). **c** Area plots of cumulative temporal dynamics of changes in the phosphoproteome (teal areas and solid lines), transcriptome (brown area and dotted lines) and proteome (purple areas and dashed lines). **d** Tile plot of the top enriched GO terms per stimulation: IFNγ (pink), TNFα (orange), TNFα + IFNγ (green) and omics level (as indicated). Color gradient indicated –log10 BH-adjusted *p* values. MF molecular function, CC cellular component, BP biological process. **e** Line plots of phosphorylation events, transcript levels and relative protein abundances of members of highly enriched GO:terms. Circles indicate medians; error bars show standard deviations (*n* = 3 biological replicates).

correlating transcripts *NFKBIE* ("TNFα"), *SECTM1* ("IFNγ"), *PARP10* ("common") and *CCL8* ("TNFα + IFNγ") (Fig. 5c). Next, we connected regulated features by querying the STRING database for high confidence (>0.95) interactions. This resulted in a network containing 2306 interactions, revealing 9 high-density hubs summarized in biological processes: "*Viral sensors*", "*Cytokines*", "*Complement factors*", "*JAK/STAT signaling*", "*Cell cycle*", "*NFKB signaling*", "*Proteasome*", "*NFKB complex*" and "*Antigen presentation*" (Fig. 5d and Supplementary Fig. 6). Plotting the ratio of response classifications per hub, only two were majorly TNFα induced: NF-κB complex proteins (47% TNFα) and cytokines (44% TNFα), while all others were primarily IFNγ-induced. Especially the hubs, "*Complement factors*", "*Viral sensors*",

"*Proteasome*" and "*Antigen presentation*" were predominantly IFNγ-induced (>75%). None of the hubs were majorly synergistically induced, suggesting synergy is confined to specific proteins and not entire biological processes.

**IFNγ-induced immune repertoire of ECs.** To delineate biological processes from this multi-omics integration, we dissected the IFNγ induced processes, as these covered mostly immune mediating processes. The "viral sensors" hub contains innate antiviral proteins *IFIT2, IFIT3, IFIH1, DDX58* and *GPB1 and GBP2* which were all induced by IFNγ (Fig. 6a). Interestingly, although protein changes generally occur at later timepoints and induced lower log-fold changes than mRNA, *DDX58* is an

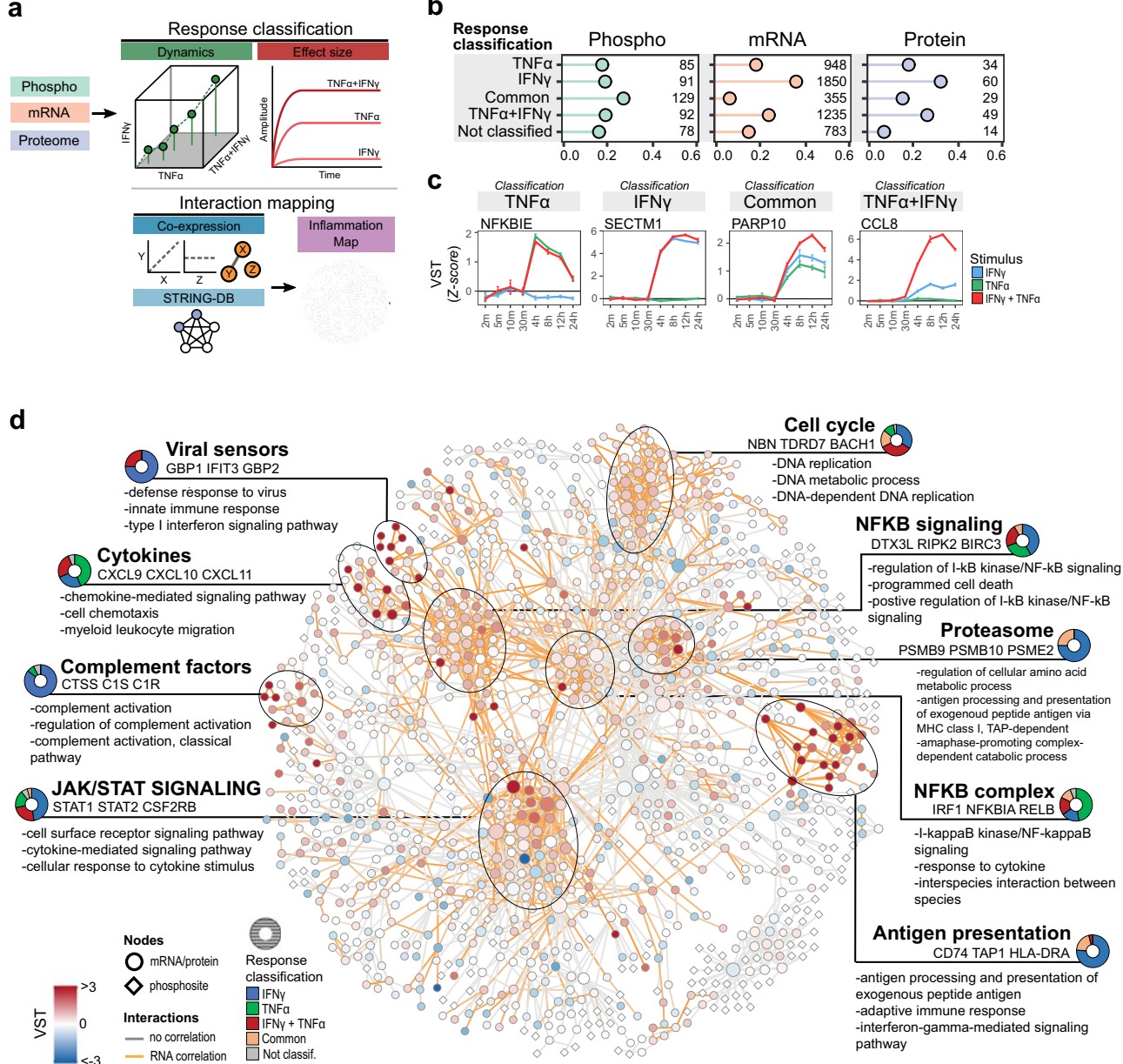

**Fig. 5 Interaction network of differentially regulated RNA/protein and phosphosites. a** Schematic overview of generating the interaction network from significant events. **b** Bar plot showing ratio of response classification and total nodes per classification and omics levels (as indicated). **c** Line plots of normalized transcript levels (VST) of highly correlating classification transcripts. Colors indicate stimulus: TNFα (green), IFNγ (blue), TNFα + IFNγ (red). Circles indicate medians of replicates; error bars show standard deviation (n = 3 biological replicates). **d** Interaction network of all differentially regulated mRNAs/proteins and phosphosites, filtered by high confidence interactions, nodes represent transcripts/proteins and phosphosites. Color gradient indicates median transcript levels at 24 h TNFα + IFNγ stimulation (n = 3 biological replicates). Per cluster are indicated: summarizing biological process, top three regulated transcripts and enriched biological processes. Donut plot indicates ratio of response classification in each cluster: TNFα (green), IFNγ (blue), TNFα + IFNγ (red), common (orange) and no classification (gray).

example of limited fold changes in transcripts (max. 1-fold), while protein increases over 3-fold. Complement factors were another strongly induced IFNγ hub, especially *C1R* and *C1S* showed drastically increased upregulation of transcripts (>5-fold) (Fig. 6b). *C3*, crucial in the activation of the alternative pathway, is the only uniquely TNFα-induced transcript in this hub. However, whether transcript expression translated to protein increases is unclear as corresponding proteins were not detected. IFNγ also induced a strong antigen-presenting hub (Fig. 6c). We previously reported TNFα induces MHCI proteins, including *HLA-A, HLA-B* and *HLA-C*, which we observed here too[36].

However, these MHCI complex proteins as well as immunoproteasome (*PSMB8, PSMB9* and *PSMB10*) and immunoproteasome regulator subunits (*PSME1* and *PSME2*), peptide loading proteins (*TAP1, TAP2, ERAP1* and *ERAP2*)[37,38] and immune checkpoint protein Programmed death- ligand 1 (*CD274*) were higher induced by IFNγ compared to TNFα. Moreover, IFNγ also induced MHCII complexes required for exogenous antigen presentation. *HLA-DR, HLA-DQ* and *HLA-DP* transcripts were upregulated 4–7-fold at 12–24 h of IFNγ stimulation. Interestingly, in contrast to MHCI proteins, which were detected abundantly on the protein level, we were only able to detect *HLA-DRA*

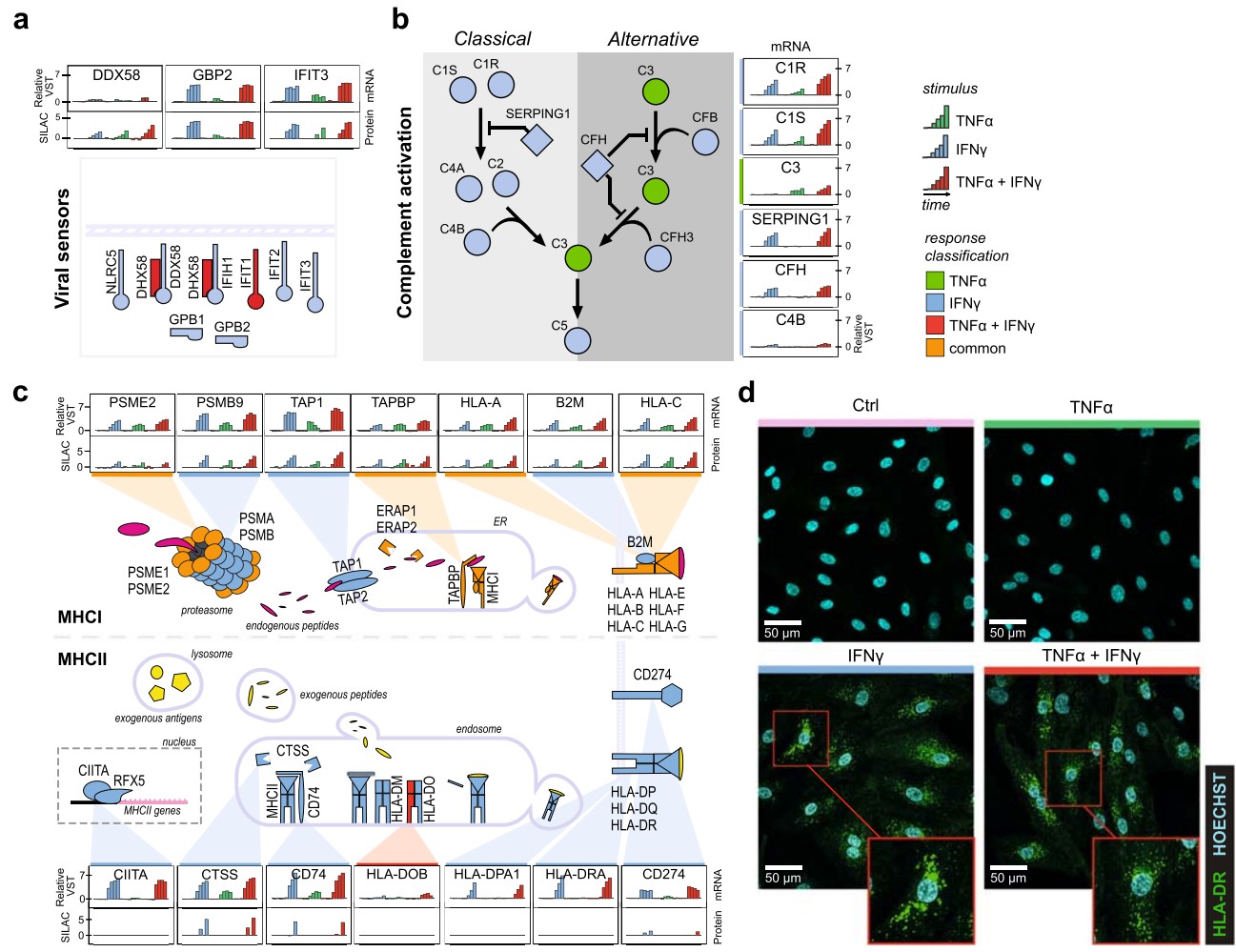

**Fig. 6 IFNγ induces regulation of viral sensors, complement factors and antigen presentation in ECs. a** Overview of regulated innate immune sensors in the "Viral Sensors" hub, **b** the "Complement factors" hub and **c** the "Antigen presentation" hub. Bar plots of highlighted transcripts/proteins indicate median transcript level (VST) or protein SILAC ratio per stimulus and timepoint ($n = 3$ biological replicates). Colors indicate stimulus: TNFα (green), IFNγ (blue), TNFα + IFNγ (red). Node fill indicates response classification of transcript: TNFα (green), IFNγ (blue), TNFα + IFNγ (red), common (orange) and no classification (gray). **d** Confocal images of HLA-DR immunostaining in unstimulated ECs (Ctrl, pink) and stimulated with TNFα, IFNγ, and TNFα + IFNγ. HLA-DR staining is depicted in green, Hoechst staining in cyan. Representative experiment shown ($n = 3$ biologically independent experiments). Upper limit of the display range were adjusted equally across images for visualization purposes.

and *HLA-DRB* in separate LFQ workflow experiments (Supplementary Fig. 7a). To visualize the discrepancy between MHCI and MHCII protein expression, we stained BOECs for *HLA-A/B/C* or *HLA-DR* after stimulation of TNFα, IFNγ or combined stimulation. MHCI showed a clear distribution over the cell membrane, also in steady-state condition (Supplementary Fig. 7b) and in line with both transcriptome and protein data, *HLA-DR* was only observed in IFNγ stimulated conditions. However, in contrast to the membrane distribution of *HLA-A/B/C*, *HLA-DR* was mostly localized to compartments inside the cell (Fig. 6d).

**Transcription factor networks at the basis of inflammatory states**. Next, we investigated whether regulation between the two main signaling axes, NF-κB and JAK/STAT, could be at the basis of the observed inflammatory states. As expected, members of the NKFB complex such as *NFKB2* (P100), *NFKB1* (P40) and *RELB* were classified as "TNFα", while key mediators of the JAK/STAT pathway, *JAK1, JAK2, STAT1, STAT2* and *STAT3* all classified as "IFNγ" (Fig. 7a). Interestingly, transcripts of IFNγ receptors *IFNGR1* and *IFNGR2* were induced by TNFα, while the TNFα receptors

*TNFR1* and *TNFR2* in the NF-κB signaling axis were induced by IFNγ. Plotting the effect size of each stimulus as a ratio of the total observed combinatorial response reveals both signaling cascades showed diverse response classifications and transcripts were not exclusively regulated by one cytokine (Fig. 7b).

In a range of transcripts, the cumulative effect size of TNFα and IFNγ was not equal to the total response of combined stimulation, indicating a synergistic relation. Among these were transcription factors *RELA, STAT5A and STAT6, JAK3*, a central kinase in IFN signal transduction[39], and *LCP2*, involved in T-cell antigen receptor-mediated signaling[40] (Fig. 7c). We also observed synergetic downregulation of *PECAM1*, a crucial molecule in maintaining EC cell junctions[41]. The synergetic regulation of transcripts did not translate to the same extent into protein levels. Only at the 24 h timepoint did correlating protein abundances in the TNFα + IFNγ condition exceed the cumulative abundance of both cytokine stimuli separately.

**Endothelial inflammatory states induce distinct secretomes**. Cytokine release by ECs is a direct avenue of immune modulation

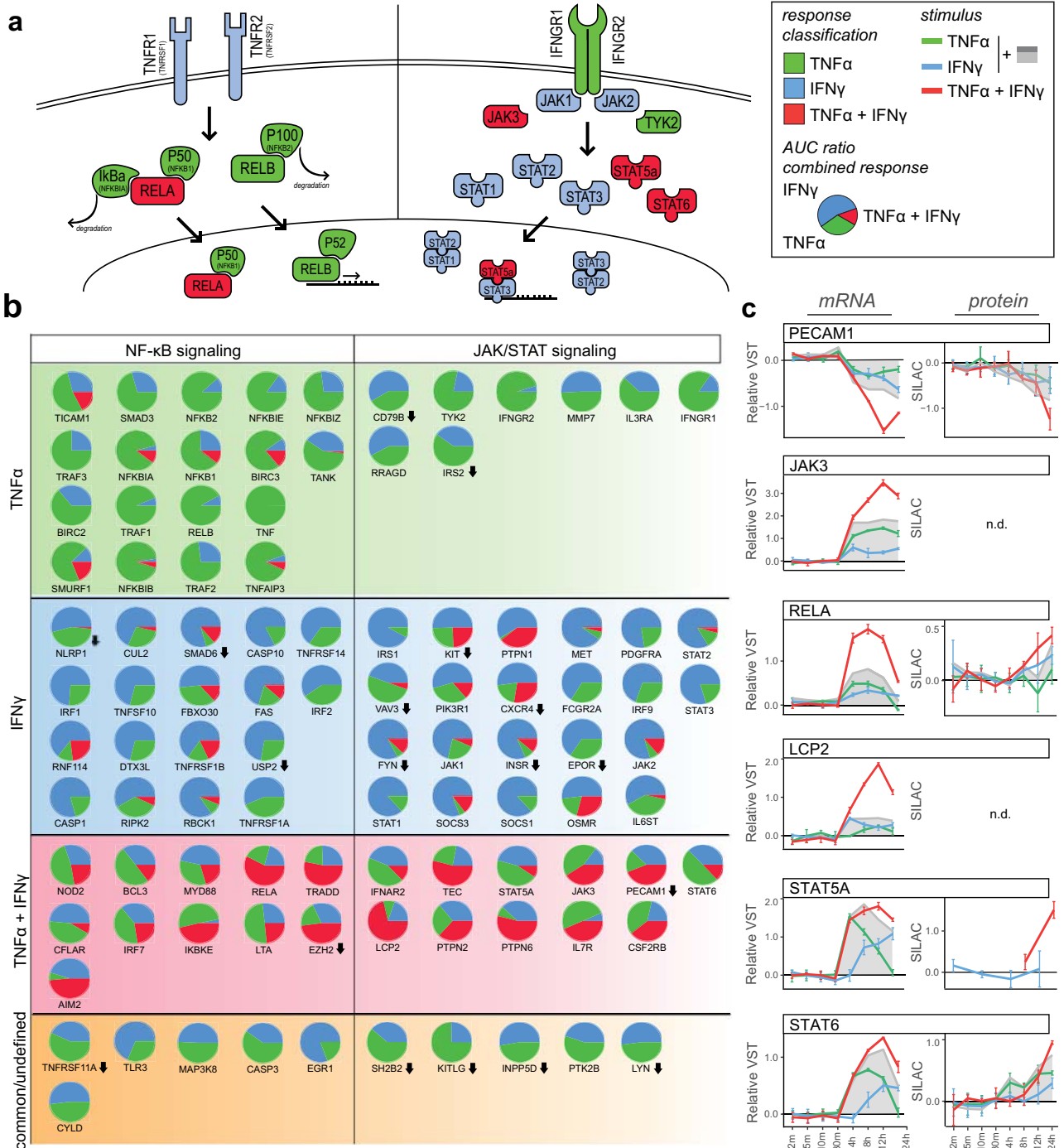

**Fig. 7 TNFα and IFNγ synergy in transcription factor signaling pathways. a** Schematic overview of NF-κB and JAK/STAT signaling axes. Node fill color indicates response classification of transcripts: TNFα (green), IFNγ (blue), TNFα + IFNγ (red), common (orange) and no classification (gray). **b** Pie charts of transcript AUC as part of the total TNFα + IFN response: TNFα (green), IFNγ (blue), [TNFα + IFNγ] - TNFα – IFNγ (red). **c** Line plots of synergistic transcript levels and SILAC ratios of correlating proteins after TNFα, IFNγ and TNFα + IFNγ stimulation. The sum of separate TNFα and IFNγ regulation is shown in gray. Circles indicate medians of replicates; error bars show standard deviation (*n* = 3 biological replicates).

by interacting with different immune cells and both TNFα and IFNγ induced distinct cytokine transcripts and synergetic increases. However, the majority of cytokines was not detected on the protein level, potentially because of the low abundance and secretory nature of cytokines[42]. Therefore, we performed secretomics experiments, following the workflow as described by Deshmukh et al.[43]. Both secretomes and cell lysates were analyzed using high-resolution LFQ MS (Fig. 8a). Time-dependent changes were most apparent in the secretome as proteins were excreted

into the supernatant over time (Supplementary Fig. 8) and we observed limited protein changes due to stimulation at early timepoints (30 m and 4 h). However, after 24 h both TNFα and IFNγ induced 29 and 54 significant proteins, respectively, and this effect was amplified in the combined condition (177 proteins) (Fig. 8b). Generating a STRING interaction network of upregulated proteins in the combined stimulation revealed 6 protein hubs enriching for extracellular proteins, such as extracellular matrix, complement proteins and cytokines/chemokines (Fig. 8c).

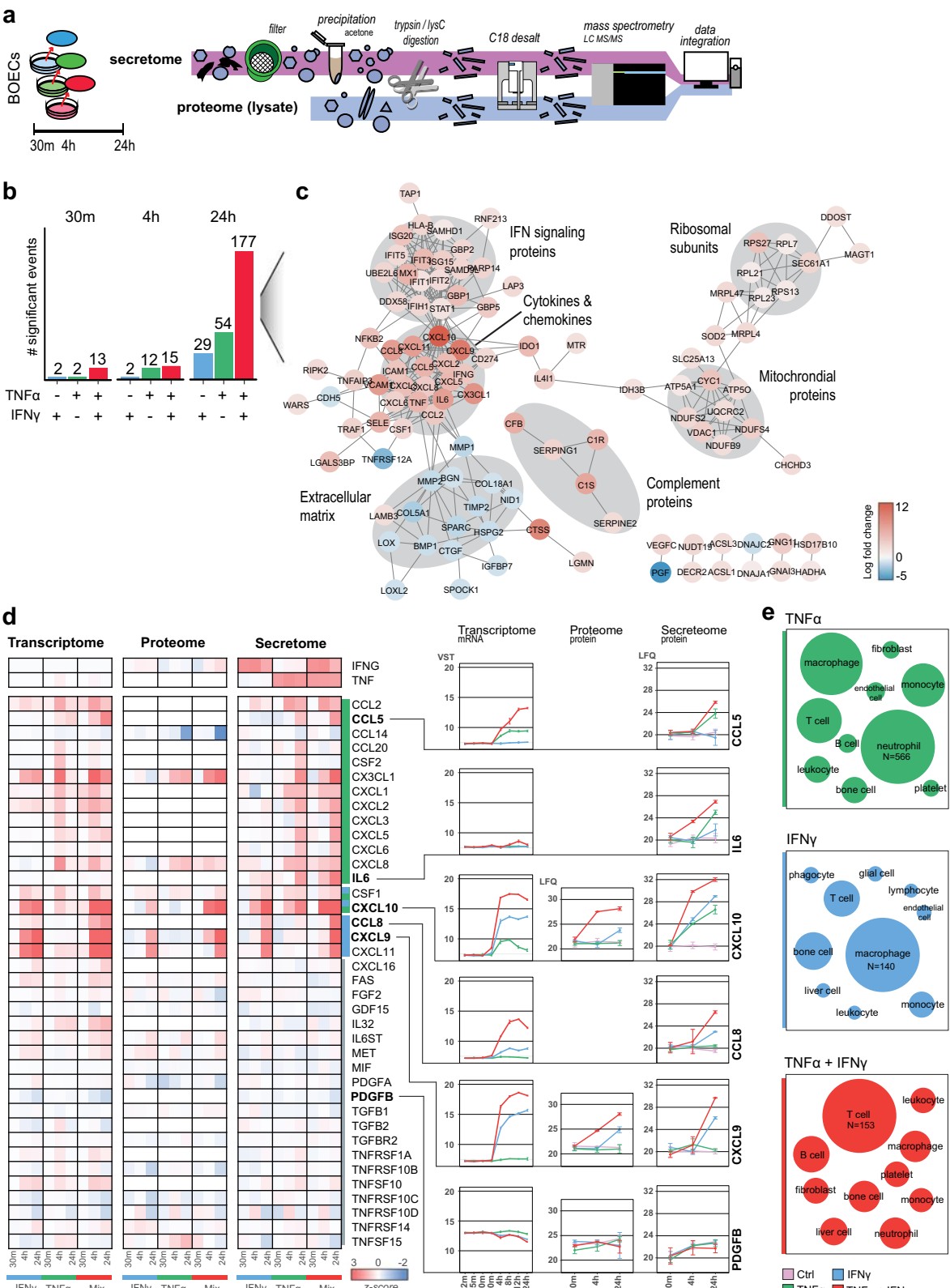

**Fig. 8 Secretome analysis of ECs after TNFα, IFNγ and TNFα + IFNγ stimulation. a** Schematic overview of secretomics workflow. **b** Bar plot of number of significantly regulated proteins per stimulus (moderated t-test, BH-adjusted p < 0.01 and log2 fold change > 1). Colors indicate stimulus: TNFα (green), IFNγ (blue), TNFα + IFNγ (red). **c** Interaction network of differentially regulated proteins after 24 h TNFα + IFNγ stimulation showing protein type per hub indicated in gray. **d** Heatmap of enriched proteins in the cytokine registry per omics level showing correlating transcripts and proteins in lysate after TNFα, IFNγ and TNFα + IFNγ stimulation. Color gradient indicates z-scores. Several proteins are highlighted in line plots showing VST and LFQ values, error bars show standard deviation (n = 3 biological replicates). **e** Number of papers enriching for cell type interactions by cytokines induced per stimulation. Node size represents number of papers, per stimulus largest node is set to most cited cell type (TNFα: n citations = 566, IFNγ: n = 140, TNFα + IFNγ: n = 153).

Two hubs contained intracellular proteins namely ribosomal units and mitochondrial proteins, indicating increased cell death. Next, we used the ImmPort Cytokine Registry to create an overview of all secreted cytokines in this experiment ($n = 44$) and plotted the intracellular protein abundance and transcriptome data (Fig. 8d). In line with the above-described inflammatory signatures, we observed distinct cytokines secreted per stimulus (13 for TNFα and 3 for IFNγ), cytokines excreted equally by both stimuli (*CSF1* and *CXCL10*) and 21 cytokines unaffected by a stimulus and constitutively expressed (e.g., *PDGFB*). The synergistic induction of *CCL5, CCL8, CXCL9* and *IL6* transcripts translated to increased secreted protein levels, but the fold-changes were less pronounced in the latter. Interestingly, whereas *IL6* showed minor mRNA increases it was abundantly present in the secretome. To gain an insight into the putative-affected cell types, we employed the ImmunXpresso text mining database and plotted the number of citations describing cytokine-cell interactions per released cytokine subset. For TNFα induced cytokines most citations described interactions with neutrophils ($n = 566$), followed by macrophages and T-cells, while IFNγ-released cytokines were cited mostly for macrophages ($n = 140$), bone (marrow) cells and T-cells (Fig. 8e). Synergistically released cytokines (*CCL5, CCL8, CXCL9* and *IL6*) enriched for papers on T-cells ($n = 153$), B-cells and bone (marrow) cell interactions, suggesting that each EC inflammatory state favors interactions with different immune cell types.

## Discussion

ECs are at the crossroads of inflammation and hemostasis and are increasingly recognized for their immunomodulatory role. We demonstrate TNFα and IFNγ induce system-wide adaptive EC inflammatory states. Moreover, the combination of TNFα and IFNγ induced a synergistic EC response. This study provides an in-depth molecular mapping on multiple regulatory levels and offers an extensive resource on the underlying regulation of endothelial inflammation.

As is well established, ECs contribute to immune cell migration directly through TNFα induced upregulation of *VCAM1, ICAM1* and *SELE*[28,36,44]. Here, we show that ECs, in addition to TNFα, can sense a plethora of cytokines and react on the proteome level, but the response remains limited for most. Moreover, we observe a large overlap between different stimuli, especially the TNFα cluster shared responses with both IL1-α and IL1-β, in accordance with previous reports[36,45]. The two other overlapping clusters contained mainly chemokines which could explain the relatively limited proteomic response. Some of these, such as MIP3a (*CCL20*) and MIP5 (*CCL15*), have been implicated in transmigratory processes of monocytes and dendritic cells[46,47], which could indicate these chemokines affect specific transmigratory processes instead of inducing cell-wide proteomic changes.

IFNγ stimulation resulted in the second-highest response, which did not overlap with any of the other cytokines and highlighted the immunomodulating capacities of ECs. IFNγ induced a strong upregulation of antiviral proteins and complement proteins. Of the latter, we observed *C1R, C1S* and *CFB* secretion into the extracellular matrix, showing the contribution of ECs to the regulation of the complement cascade. As expected, IFNγ also induced a strong increase of MHCII complex proteins. ECs are implicated as modulatory antigen-presenting cells, regulating CD8+ and CD4+ T cell tolerance and transmigration and play a role in allograft rejection[48–50]. Interestingly, although MHCII is constitutively expressed in vivo, our confocal analysis revealed that after 24 h of simulation, *HLA-DR* proteins were not expressed at the cell surface but were mostly confined to

intracellular compartments. This could be an artifact of in vitro culturing ablating the transport of *HLA-DR* proteins to the cell surface and potentially longer stimulation windows or a secondary trigger is necessary to facilitate cell surface expression[51,52]. In addition to the internal molecular mechanisms, another avenue of endothelial immunomodulation is through extracellular communication in the form of released signaling proteins. In line with intracellular processes, different proteins were secreted depending on the stimulus. TNFα is well studied in the context of neutrophil transmigration and cytokines released after TNFα stimulation did indeed enrich mostly for papers on neutrophil interactions. IFNγ stimulation induced secretion of *CXCL9, CXCL10* and *CXCL11*, which all signal via the CXCR3 receptor that is mostly present on monocytes, T-cells, NK-cells, and dendritic cells[53,54]. However, TNFα does not exclusively induce granulocyte attractants (e.g., *CCL20* and *CX3CL1*) and most chemokines have chemoattractant properties for multiple leukocytes[55]. Combined TNFα and IFNγ synergistically increased *CCL5, CCL8, CXCL9* and *IL6* levels, further nuancing which immune cells are favored by the endothelium. Moreover, *IL6* is associated with inflammatory disorders and implicated as a marker for disease severity in for example ARDS and COVID-19[56,57].

Comparing the temporal nature of TNFα and IFNγ responses showcased different dynamics between mRNA and proteome, while both cytokines induced a similar immediate phospho-signaling response. On the transcriptome level, TNFα induced a more immediate response which generally decreased within 24 h. Interestingly, TNFα-induced expression of surface adhesion molecules such as *VCAM1*, can last for several days, indicating a small induction can induce lasting protein expression[58,59]. On the contrary, IFNγ induced mRNA expression increases steadily over time reaching peak transcript levels at 24 h, the endpoint in this study. Whether the IFNγ response is sustained remains to be elucidated, but studies have shown a lasting proteome response for 48 h after short-term (3 h) IFNγ simulation windows[60].

The synergetic effects of combined TNFα and IFNγ stimulation in ECs have been observed in previous reports[21–23] and in line with these observations, combining both TNFα and IFNγ resulted in system-wide cooperative regulation as well. However, protein induction was generally less pronounced compared to synergetic mRNA increases and changes in protein abundance occurred mainly between 12 and 24 h, which could indicate synergy in the proteome is more apparent at later timepoints. The molecular basis of the synergetic interplay can be multifold. One mechanism could be an increased mRNA expression by both cytokines. For example, *CXCL10* is reported to have two transcription factor binding regions, one for NF-κB and one for interferon sensitive response element[22,61] and we indeed observe combined stimulation to be a summation of both separate stimuli. Other studies have described altered mRNA half-life times through cytokine stimulation[62]. For example, *CCL5* mRNA can be stabilized through IFNγ stimulation, and we indeed observe a synergetic increase in mRNA levels when combined with TNFα even though IFNγ alone does not induce transcript expression[63,64]. In contrast, the synergetic decrease of *PECAM1* transcripts could be explained by cytokine-induced mRNA destabilization[65]. Moreover, these mechanisms are specifically regulated per protein. For example, *CCL5* and *CCL8*, both synergistically induced proteins, show opposite induction patterns: for *CCL5* TNFα induces, and IFNγ stabilizes mRNA, while for *CCL8* these roles are reversed.

The activation of different NF-κB and STAT transcription factors seems to underly the different inflammatory states and we observe crosstalk in the two main activation pathways of these transcription factors. Especially synergetic induction of key mediators of these signaling axes such as NF-κB complex protein

*RELA* and *JAK3* and *STAT5A/6* suggests the differential activation of transcription programs through combined stimulation. Considering interactions between transcription factors such as NF-κB, interferon regulatory factors and STATs[22,66], this drastically increases the complexity of regulation driving inflammatory states. When putting our findings in the context of in vivo regulation we should consider the limitations of this study. We employ BOECs as our in vitro model, which are less well-characterized than other used EC models such as HUVECs. However, BOECs have been shown to express mature EC markers over multiple passages, are preferred for metabolic labeling strategies due to their extensive proliferative ability and can be directly derived from adult donors and patients[24,25,67–69]. Variability in donor-to-donor responses is a point of concern as well[70,71], but throughout this study, we employed BOECs derived from 19 different donors (Supplementary Table 2) and observed consistent inflammatory signatures throughout. Growth conditions of cultured ECs such as used matrix, 3D culturing, or flow could also affect inflammatory outcome. However, although ECs alter their cellular organization and phenotypical characteristics, proteins induced upon inflammatory triggers seem conserved between culture conditions, suggesting these additions nuance the inflammatory response instead of alternating the activated processes[72,73]. Ideally, an in vitro model combining flow and 3D culturing would be employed, but these models often lack robustness and throughput. ECs are also heterogenous between different organs and vessel-types[74] and it is unclear how inflammatory processes are regulated throughout ECs from different vascular beds. An important distinction considering vascular inflammatory disorders, is that between the micro- and microvasculature. Although studies have shown ECs derived from the macro- or microvasculature retain their specific differences in culture[75], it is difficult to assess whether BOECs take on a macro- or micro-like vessel type as they differentiate within the in vitro microenvironment. Comparing steady-state mRNA data highlighted the challenges of assessing EC variation over multiple studies. We did observe overall similar relative expression levels of EC markers and TNFα and IFNγ receptors, suggesting the observed cytokine responses could potentially be translated to other EC types as well[30–32]. To assess whether these inflammatory states are conserved between different vascular beds is a crucial step in future studies to understand tissue-specific EC inflammation. In conclusion, this study provides a detailed insight of the inflammatory states of the endothelium which is regulated through intricate transcriptional and translational control. Uncovering these molecular mechanisms is vital in understanding the paths that lead to endothelial dysfunction and its contribution to vascular inflammation.

## Methods

**Cell culture**. BOECs were isolated from healthy donors as described by Ramirez et al.[24]. For all experiments, except the multi-omics and secretome experiments, three different pools of three unique BOEC donors (mixed sexes and ages) were used (Supplementary Table 2). For the multi-omics and secretome experiments, BOECs from three different donors (mixed sexes and ages) were pooled. Culture flasks and dishes were coated with collagen type I (50 μg/ml, BD biosciences) for 1 h prior to use. Cells were cultured in EC basal medium (Lonza) supplemented with 18% FCS (Bodinco) and EGM bulletkit (Lonza) unless stated otherwise. For SILAC labeling, BOECs were maintained for 5 passages as described by Beguin et al.[36] in custom-made EGM medium (Lonza), containing EBM2 medium (not containing Arganine and Lysine) (Lonza), EGM bulletkit (Lonza) and 18% FCS for passages 1–3 and in 18% 1 kDa dialyzed FCS for passages 4–5. Cells were SILAC labeled by the addition of isotope-labeled amino acids during all passages (light: Arg0 and Lys0, medium: Arg6 and Lys4, heavy: Arg10 and Lys8, Cambridge Isotopes). After five passage incorporation of labeled amino acids reached >95% in the total proteome.

**Stimulation**. All recombinant human cytokines used for stimulations were obtained from Peprotech (Supplementary Table 1). ECs were stimulated in three biological replicates with 10 ng/ml per cytokine for indicated timepoints, with the exception of dose-response experiments, in which cells were stimulated at 1, 10 and 100 ng/ml. Prior to stimulation, cells were washed 3× with PBS and stimulations were performed in endothelial basal medium (Lonza) supplemented with 18% FCS (Bodinco) and EGM bulletkit (Lonza), with the exception of SILAC BOECs and secretome experiments. SILAC BOECs were serum starved for 2 h prior to stimulation and stimulated in endothelial basal medium (Lonza) without additions. Stimulations in secretome experiments were performed in phenol-red-free endothelial basal medium (Promocell) without any additions.

**Immunofluorescence staining**. BOECs were grown to confluence on collagen-coated glass coverslips. After 4 days, cells were either not stimulated or stimulated with, TNFα, IFNγ or TNFα + IFNγ as described above. Cells were fixed using 4% PFA (Thermo Scientific), washed 3x with PBS and quenched with 50 mM ammonium chloride (Sigma-Aldrich). Antibody staining steps were performed in 1% BSA (Serva), 0.1% Saponin (Sigma-Aldrich) to permeabilize cells. MHCI was stained using pan-HLA monoclonal W6/32 mouse antibody generated from hybridoma (ATCC, HB-95), HLA-DR was stained with monoclonal L243 anti-human/monkey antibody (InVivoMAb, BE0306), both at 10 μg/ml. Alexa Fluor 488 chicken-anti-mouse conjugated secondary antibody (2 μg/ml) was used for both stainings (Invitrogen, #A21200). Slides were fixed in Mowiol 4-88 (Polysciences). Pictures were taken on an SP8 Confocal Laser Scanning Microscope (Leica) with a 40×/1.30 oil objective (Leica, 11506359) at 1024 × 1024 resolution. Images were processed using Fiji[76]. Immunostaining was performed three times in independent experiments.

**RNA sequencing**. Cells used for RNAseq analysis were lysed in RLT buffer (QIAGEN) according to the manufacturer's protocol. RNA sequencing was performed by GeneWiz (Azenta life sciences), including RNA isolation, library preparation, strand-specific RNAseq with PolyA selection and Illumina paired-end 150 bp sequencing. After quality control with FastQC, sequences were aligned to the human ChGR38.104 genome reference using STAR 2.7.8a and reads were summarized using featureCounts 2.0.1. Differential expression analysis was performed using DESeq2[77], applying a significance threshold of a Benjamini–Hochberg (BH) multiple testing corrected $p$ value of <0.05 and log2 fold change of >1.

**Mass spectrometric analysis**. For mass spectrometry analysis of EC proteomes, cells were lysed in 1% sodium deoxycholate (Bioworld), 10 mM TCEP (Thermo Scientific), 40 mM chloroacetamide (Sigma-Aldrich), 100 mM Tris-HCl pH 8.0 (Gibco) supplemented with 1× HALT protease/phosphatase inhibitor (Thermo Scientific). Lysates were incubated for 5 min at 95 °C and sonicated for 10 min in a sonifier bath (Branson model 2510), after which trypsin (Promega) was added in a 1:50 (w/w) protein ratio. Peptides were desalted with C18 cartridges (Agilent) according to manufacturers' instructions and where applicable phosphopeptide enrichment was performed using Fe(III)-IMAC cartridges (Agilent) as described by Post et al.[78] on an AssayMAP BRAVO (Agilent). For the deep proteome protein was first cleaned with HyperSep C18 Cartridges (Thermo Scientific) and HyperSep Hypercarb SPE Cartridges (Thermo Scientific). Then, samples were fractionated with the Pierce High pH reversed-phase peptide fractionation kit (Thermo Scientific) and fractions desalted with Empore C18 STAGE tips (Supelco). Fractionation was performed in triplo and obtained fractions were measured separately. For secretome analysis, samples were worked up as described in Deshmukh et al.[43] using minor adjustments. First, supernatants were collected and filtered using a 0.2-μm filter (Whatman). Collected supernatants were spun down at 5000 g to remove cell debris and stored at −80 °C before further use. Samples were acetone (1:4 ratio, Biosolve) precipitated overnight at −20 °C, before precipitates were lysed with Urea (6 M, Invitrogen) + ThioUrea (2 M, Sigma-Aldrich), reduced with 10 mM DTT (40 min, Thermo Scientific), and alkylated with 55 mM IAA (Thermo Scientific) in the dark (40 min). Samples were subsequently digested with 0.5 ug LysC/Trypsin (Thermo Scientific) overnight and desalted on Empore C18 STAGE tips (Supelco).

Peptides were separated by nanoscale C18 reverse chromatography coupled online to an Orbitrap Fusion Lumos Tribrid mass spectrometer or Orbitrap Fusion Tribrid mass spectrometer (Thermo Fisher Scientific) via a nanoelectrospray ion source at 2.15 kV. Buffer A was composed of 0.1% formic acid and buffer B of 0.1% formic acid and 80% acetonitrile. For label-free analysis, peptides were loaded for 17 min at 300 nl/min at 5% buffer B, equilibrated for 5 min at 5% buffer B (17–22 min) and eluted by increasing buffer B from 5 to 27.5% (22–122 min) and 27.5 to 40% (122–132 min), followed by a 5 min wash to 95% and a 6 min regeneration to 5%. Survey scans of peptide precursors from 375 to 1500 m/z were performed at 120,000 resolution (at 200 m/z) with a $4 \times 10^5$ ion count target. Tandem mass spectrometry was performed by isolation with the quadrupole, with isolation window 0.7, higher energy collisional dissociation (HCD) fragmentation with normalized collision energy of 30 and rapid scan mass spectrometry analysis in the ion trap. The tandem mass spectrometry (MS2) ion count target was set to $3 \times 10^4$, and the max injection time was 20 ms. Only those precursors with charge state 2–7 were sampled for MS2. The dynamic exclusion duration was set to 30 s with a 10 ppm tolerance around the selected precursor and its isotopes.

Monoisotopic precursor selection was turned on. The instrument was run in top speed mode with 3 s cycles. All data were acquired with Xcalibur software (Thermo Fisher Scientific).

For phosphoproteomics acquisition of SILAC labeled samples, tryptic peptides were loaded for 17 min at 300 nl/min at 5% buffer B, equilibrated for 5 min at 5% buffer B (17–22 min) and eluted by increasing buffer B from 5 to 15% (22–87 min) and 15 to 38% (87–147 min), followed by a 10 min wash to 90% and a 5 min regeneration to 5%. Survey scans of peptide precursors from 350 to 1750 m/z were performed at 240 K resolution (at 200 m/z) with a $2 \times 10^5$ ion count target. Tandem mass spectrometry was performed by isolation with the quadrupole with isolation window 1.6, HCD fragmentation with normalized collision energy of 30, and rapid scan mass spectrometry analysis in the orbitrap. The MS2 ion count target was set to $10^5$ and the max injection time was 60 ms. Only those precursors with charge state 2–7 were sampled for MS2. The dynamic exclusion duration was set to 60 s with a 10-ppm tolerance around the selected precursor and its isotopes. Monoisotopic precursor selection was turned on. The instrument was run in top N mode. For SILAC proteome samples a slightly adjusted protocol was used: peptides were loaded for 17 min at 300 nl/min at 5% buffer B, equilibrated for 5 min at 5% buffer B (17–22 min) and eluted by increasing buffer B from 5 to 28% (22–80 min) and 28 to 40% (80–85 min), followed by a 5 min wash to 95% and a 5 min regeneration to 5%. Survey scan was set to 240 K with a $1 \times 10^6$ ion count target. Tandem mass spectrometry was performed on the 10 most intense ions by isolation using the quadrupole and analysis in the ion trap at a resolution of 30 K. The MS2 ion count target was set to $5 \times 10^4$ with a maximum injection time of 60 ms. The instrument was run in top speed mode with 3 s cycles. All data were acquired with Xcalibur software.

**Mass spectrometry data analysis**. The RAW mass spectrometry files were processed with the MaxQuant computational platform, 1.6.2.10. Proteins and peptides were identified using the Andromeda search engine by querying the human Uniprot database (release 2019). Standard settings with the additional options match between runs, LFQ, IBAQ, and unique peptides for quantification were selected. For SILAC samples, multiplicity was set to 3 (for Arg0 and Lys0, Arg6 and Lys4 and Arg10 and Lys8) and the re-quantify option was enabled, where applicable Phospho STY was set as a dynamic modification. Data were analyzed using R 3.5.2/RStudio 1.1.456. For label-free data, "reverse", "potential contaminants" and "only identified by site" peptides were filtered out. Proteins and phosphosites were filtered for at least 100% valid values per experimental group. LFQ values were transformed in $\log_2$ scale. Missing values were imputed by a normal distribution (width = 0.3, shift = 1.8), assuming these proteins were close to the detection limit. Batch effects were corrected for using ComBat in the sva package. Label-free statistical analyses were performed using LIMMA[79]. For SILAC data analysis, statistical analysis was performed using a linear model without intercepting non-imputed data. For clustering purposes, missing values were imputed by linear approximation using the Amelia package. For both label-free and SILAC data, moderated t-tests were used to determine differentially abundant proteins[80]. A BH-adjusted $p < 0.05$ and log2 fold change > 1 was considered significant and relevant. For label-free secretomics data, a BH-adjusted $p < 0.01$ and log2 fold change > 1 was used as the significance threshold.

**Mapping cytokine-endothelial interactions**. Receptor and ligand definition was performed as described by Rieckmann et al.[8]. In brief, "extracellular" keywords or "GPI-anchor" topology domain based on the Uniprot knowledge base annotations were used to define receptor proteins. STRING-DB interactions with known interacting proteins (combined scores >0.4) were selected and annotated for Uniprot "secreted" and "signal" keywords or GO:CC "extracellular space" and "extracellular region" terms to define receptor ligands. Connections between receptors and ligands were visualized in Cytoscape 3.8.0.

**Response classification and inflammation map construction**. RNA, protein and phosphosite responses in the TNFα + IFNγ stimulation were classified based on a combined assessment of Pearson correlation with single TNFα and IFNγ conditions and effect size by determining areas under the curves using the flux packages. First dynamics were categorized as either S1-TNFα shape (correlation coefficient >0.7 with TNFα and <0.7 with IFNγ stimulation); S2-IFNγ shape (correlation coefficient <0.7 with TNFα and >0.7 with IFNγ stimulation); S3-common shape (correlation coefficient >0.7 with TNFα and >0.7 with IFNγ stimulation) or S4-TNFα + IFNγ shape (correlation coefficient <0.3 with TNFα and <0.3 with IFNγ stimulation). Effect sizes were categorized as E1-TNFα effect (AUC ratios TNFα + IFNγ stimulation/TNFα stimulation <2 and TNFα + IFNγ stimulation/IFNγ stimulation >2); E2-IFNγ effect (AUC ratios TNFα + IFNγ stimulation/TNFα stimulation >2 and TNFα + IFNγ stimulation/IFNγ stimulation <2), and E3-TNFα + IFNγ effect (AUC ratios TNFα + IFNγ stimulation/TNFα stimulation >2 and TNFα + IFNγ stimulation/IFNγ stimulation >2). Classifications were set to common if S3 but not E3 criteria were fulfilled; TNFα classification: S1 or S3 + E1; IFNγ classification: S2 or S3 + E2; TNFα + IFNγ classification: S3 or E3 classifications were fulfilled. If not fulfilling any shape or effect size cutoffs, classification was set to "not classified".

To construct an inflammation map, all transcript, proteins and phosphosites that were statically significant in the TNFα + IFNγ stimulation were selected and collapsed to gene names. Based on STRING-DB interaction scores >0.9 gene names were connected with edges. Edges between phosphosites, corresponding proteins and transcript were manually appended to the network. This network was visualized in Cytoscape 3.8.0. We first obtained the "EdgeBetweenness" using the "Analyse Network" function, after which we used "Edge-weighted Spring Embedded Layout" to visualize the network. We highlighted interaction hubs based on closeness of nodes, overall regulation levels and biological overlap.

**Co-expression, term enrichment and EC comparative analyses**. Co-expression analysis was performed using the WGCNA package[81] using a signed network and a soft power of 4, minClusterSize was set to 10. GO term enrichment and pathway analyses were performed using clusterPofiler[27] and rWikiPathways[82] packages, enrichments with a BH-adjusted $p$ value <0.05 were considered significant. For the comparative EC analyses, we selected three studies with various primary cultured ECs. To compare datasets, batch effects were removed and normalized using the Limma package[79].

**Statistics and reproducibility**. Statistical tests are employed and significance cut-off values are indicated per experiment in the methods section. For reproducibility, ECs from 19 different donors were randomly combined in different pools of three donors (Supplementary Table 2). All stimulations were performed in at least three biological replicates, as indicated in per experiment.

**Reporting summary**. Further information on research design is available in the Nature Portfolio Reporting Summary linked to this article.

## Data availability

The mass spectrometry proteomics data have been deposited to the ProteomeXchange Consortium via the PRIDE[83] partner repository with the dataset identifier PXD036582. The mRNA sequencing data have been deposited in NCBI's Gene Expression Omnibus[84] and are accessible through GE Series accession number (GSE213111). Source data used for all figures in this study can be found in Supplementary Data 1–5. Any remaining information can be obtained from the corresponding author upon reasonable request.

## Code availability

In-house written scripts are available from the corresponding author upon reasonable request.

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

## Acknowledgements

This work was supported by the Landsteiner Foundation for Blood Transfusion Research grants LSBR-1517 awarded to M.v.d.B. and LSBR-1923 awarded to A.J.H. and M.v.d.B.

## Author contributions

S.A.G. performed experiments, analyzed data, and drafted the manuscript. E.R.S. analyzed data. E.F.J.J. and B.L.v.d.E. performed experiments. F.P.J.v.A. and C.v.d.Z. performed mass-spec data acquisition. A.B.M supervised research. A.J.H. designed the study, performed experiments, supervised research, analyzed data, and drafted the manuscript. M.v.d.B. designed the study, supervised research, and drafted the manuscript. All authors read and approved the final version of the manuscript.

## Competing interests

The authors declare no competing interests.
