## [Peer Review File · Communications Biology]

Reviewers' comments:

Reviewer #1 (Remarks to the Author):

Dear Authors,

in your MS you describe a multi-omics approach to investigate inflammatory stages of endothelial cells.

However, there are a few points that should be addressed.

1) Your model system, BOECs is unusual in EC research. You probably should mention the reason for choosing it as opposed to other models such as human microvascular ec.

2) Given the unusual EC model and the heterogeneity of ECs, did you confirm this repertoire on other endothelial cell types by means of e.g. transcriptomics mining in databases, for example in endoDb? If not, this should probably be done in a short addition to the paper.

3) Did you consider to replace the Term Enrichment Analysis with a more recent method, for example Gene Set Variation Analysis? This way, the somewhat arbitrarily logFC threshold could be avoided. Additionally, this approach allows to get information about the directionality of involved terms, resp. pathways.

4) Structuring of the M&Ms. As the MS is written, the distinction between wet-lab and dry-lab methods is blurred. I suggest to restructure the M&Ms as follows: cell culture, stimulation, IF staining, RNASeq, Proteomics, Differentially expressed genes and proteins, WGCNA, Mapping, and the rest. I trust you get the picture.

5) I am not sure about the relevance of the impedance testing part. If it aims at validating the model in terms of functionality, that should be briefly explained, also, this part should then be placed at the beginning of the MS.

6) The visualization of the WGCNA is a bit confusing. As I understood Figure 2C you split the dendrogram (consisting of cytokines) into 5 modules, calculated the z-score of the respective genes and showed them colour coded in the respective module strip. It would help to add a color bar showing the module annotation for the respective gene.

A final remark ME is the nomenclature for the Module Eigenene, therefore modules should be named "M".

Reviewer #2 (Remarks to the Author):

Herein I submit a review of a manuscript entitled "Multi-omics Delineation of Cytokine-Induced Endothelial Inflammatory States" authored by Groten et al.

This study represents a major effort to depict potential protein interactions involved in TNF α /INF γ -induced inflammation in endothelial cells based on the compilation of some rather impressive datasets and their interrogation. Some of these data recapitulate previous findings; however, there is sufficient novelty to warrant publication. I am very much impressed with this manuscript but I do have some concerns described below.

BOECs are an interesting choice of cells to conduct this study. Can the authors comment on whether these cells adopt a microvascular or macrovascular phenotype and include it in the discussion? If microvascular, these findings may be particularly significant to eye and kidney vasculopathies? Furthermore, could it be possible that some of the observed mRNA and protein changes under the stimuli described, be attributed to retention of EPC character?

Interestingly, ECIS experiments were conducted and the authors claimed that the BOEC-derived EC demonstrated barrier function yet only a normalized resistance was reported. What were the raw resistance readings? Typically for those EC that establish true barriers, the resistance measures are in the neighborhood of 3000 ohms. The resistance to ion flow derives from two components, one is due to cell attachment to the substratum and the other to the formation of tight junctions in the paracellular cleft. Is the relative resistance shown in figure 3b simply due to cell attachment or is a true barrier formed?? IHC to demonstrate tight -junction disorganization under the stimuli that were tested and the inclusion of FITC-dextran permeability as an additional measure of monolayer permeability would help clarify this issue.

In the methods section BOECs were cultured in 18% fetal calf serum (FCS) for the secretome analysis. I presume they switched to ECs at some point ? I also presume that the medium was changed to add the cytokine stimuli ? It would be helpful to include a few more details that may have been omitted. Along those lines, an impressive list of cytokines was detected in the conditioned medium, were these samples manipulated in some manner to deplete the 18% FCS? I checked reference 34 and it doesn't seem to suggest they were. If these samples were not depleted of this bulk protein I am surprised that you were able to detect these cytokine in the conditioned medium. Would use please clarify?

Referee expertise:

Referee #1: Network analysis, endothelial cells

Referee #2: Vascular diseases

Reviewers' comments:

Reviewer #1 (Remarks to the Author):

Dear Authors,

in your MS you describe a multi-omics approach to investigate inflammatory stages of endothelial cells.

However, there are a few points that should be addressed.

1. Your model system, BOECs is unusual in EC research. You probably should mention the reason for choosing it as opposed to other models such as human microvascular etc.

-We thank the reviewer for raising this point. We make use of BOECs as our *in vitro* model system for endothelial cells for several reasons. The metabolic labeling strategy performed here requires the superior growth capacity of BOECs. In addition, a great translational advantage of BOECs is that they can be obtained from patient blood which has previously enabled us to assess disease-specific endothelial cell responses^{1,2}. We have previously also shown that these cells maintain hallmark endothelial characteristics during culture, including cobblestone morphology, expression of endothelial-specific Weibel-Palade bodies and endothelial thrombin responses^{3,4}. We agree that our reasons for choosing this model were not clearly mentioned and have added a short comment in the introduction and reiterated these reasons for our choice of BOECs in the discussion.

[56-58, page 2, **Introduction**] "Therefore, in this study we set out to dissect the molecular signatures of endothelial-cytokine responses, employing blood outgrowth endothelial cells (BOECs) also known as Endothelial Colony Forming Cells (ECFCs) as our source of ECs because of their extensive robust expansion, expression of mature vascular EC markers ability to be isolated from adult donors and patients^{24,25}."

[355 – 359, page 12, **Discussion**] "We employ BOECs as our *in vitro* model, which are less well-characterized than other used EC models such as HUVECs. However, BOECs have been shown to express mature EC markers over multiple passages, are preferred for metabolic labeling strategies due to their extensive proliferative ability and can be directly derived from adult donors and patients^{24,25,64-66}."

2. Given the unusual EC model and the heterogeneity of ECs, did you confirm this repertoire on other endothelial cell types by means of e.g. transcriptomics mining in databases, for example in endoDb? If not, this should probably be done in a short addition to the paper.

-We thank the reviewer for this excellent suggestion and bringing endoDB and its value for transcriptomic data mining to our attention. EC heterogeneity is a very topical point in vascular inflammation and this resource provides an excellent hub to compare our finding with other studies and laboratories. We selected the 3 largest published transcriptomics studies encompassing several

different types of primary cultured ECs and BOECs put these next to our BOEC mRNA dataset. Expression of EC-markers are comparable between EC types, as well as TNF α and IFN γ receptors, suggesting this inflammatory response could be translatable to other EC-types. However, this analysis also highlighted the challenges of comparing ECs between different studies and laboratories. We have added these results as a supplemental figure (**Figure S5**) and address these in the result section and in the discussion:

[156-162, page 6, **Results**] “To assess how the steady-state repertoire of BOECs compared to other EC types we compared our RNAseq data to 3 published studies on various cultured primary ECs within the endoDB29–32. Principal component analysis showed high overlap with HUVECs in Rombouts et al. and BOECs and pulmonary ECs in Long et al. (**Figure S5A**). However, correlation of transcriptome signatures also highlighted study induced variation (**Figure S5B**). Relative expression levels of key EC genes as well as TNF α and IFN γ receptors were similar between the majority of EC types (**Figure S5C**).”

[374 – 379, page 12,13, **Discussion**] “Comparing steady state mRNA data highlighted the challenges of assessing EC variation over multiple studies. We did observe overall similar relative expression levels of EC markers and TNF α and IFN γ receptors, suggesting the observed cytokine responses could potentially be translated to other EC types as well³⁰⁻³². To assess whether these inflammatory states are conserved between different vascular beds is a crucial step in future studies to understand tissue-specific EC inflammation”.

Figure S5

Figure S5: Transcriptome comparison of steady state EC type.

A. PCA plot of EC types from different studies after batch correction. Color indicates EC type: BOEC (blood outgrowth), HAEC (aortic), HCAEC (cardiac artery), HHVEC (hepatic vein), HIAEC (iliac artery), HIVEC (iliac vein), HPAEC (pulmonary artery), HPVEC (pulmonary vein), HUAEC (umbilical vein) and HHAEC (hepatic artery).

B. Heatmap showing Pearson correlation between samples. Color gradient scale represents Pearson correlation. Samples in this study are highlighted.

C. Relative expression levels of key EC markers between EC types. Points indicate relative expression of each sample, bar indicates average (N = 3-9).

3. Did you consider to replace the Term Enrichment Analysis with a more recent method, for example Gene Set Variation Analysis? This way, the somewhat arbitrarily logFC threshold could be avoided. Additionally, this approach allows to get information about the directionality of involved terms, resp. pathways.

-We did not consider using the Gene Set Variation analysis, and we thank the reviewer for their suggestion as it has provided another tool in our analysis workflow toolbox. Following up on this comment we performed GSVA enrichment on our mRNA data using the KEGG pathway database (see the figure below). Although it does provide a robust method to distinguish a broad range of regulated pathways per stimuli, it remained difficult to pinpoint pathways of interest from the broad overview. Moreover, it deemed challenging to apply this method in SILAC based proteomics, due to the presence of missing values as SILAC does not permit imputation strategies and due to nature of the ratio determination. Furthermore, although GSVA provides a robust method to identify pathways, as with all annotation driven techniques it is limited by the comprehensiveness of the used database. For these reasons we have decided to not include the GSVA analysis in our manuscript.

Heatmap of GSVA enrichment score of top five significantly regulated pathways per stimulus and timepoint. Color gradient scale represents GSVA enrichment score per condition, n=3 replicates. Color code represents in which condition pathway is significantly up- or down-regulated compared to 0h control (p-value < 0.05).

4) Structuring of the M&Ms. As the MS is written, the distinction between wet-lab and dry-lab methods is blurred. I suggest to restructure the M&Ms as follows: cell culture, stimulation, IF

staining, RNASeq, Proteomics, Differentially expressed genes and proteins, WGCNA, Mapping, and the rest. I trust you get the picture.

-We agree with the reviewers' remark and we have restructured the M&Ms in accordance with these comments.

[388-595, page 14 – 20, **Methods**]

5) I am not sure about the relevance of the impedance testing part. If it aims at validating the model in terms of functionality, that should be briefly explained, also, this part should then be placed at the beginning of the MS.

-We agree the ECIS experiment is not directly relevant to our molecular analysis or conclusions. Also considering the comments of reviewer 2 (see also comment 2 of reviewer #2), we did not take into account the complexity and nuances of ECIS experiments and have opted to exclude the experiment (previous Figure 3B) from this study.

6) The visualization of the WGCNA is a bit confusing. As I understood Figure 2C you split the dendrogram (consisting of cytokines) into 5 modules, calculated the z-score of the respective genes and showed them colour coded in the respective module strip. It would help to add a color bar showing the module annotation for the respective gene.

-We apologize that our explanation and visual representation of this figure was unclear. To summarize, we performed WGCNA on the significantly regulated proteins, resulting in 5 modules. Next, we determined the median LFQ intensity of the proteins in a module per stimulus and subsequently scaled this over all cytokines in a Z-score. This enables us to visualize which stimuli contributed to a module. We then show hallmark proteins (with high module membership scores) induced by these stimuli per module and perform the GO:term analysis on all proteins per module to provide context of the biological process/pathway present in these modules.

Following the reviewers suggestion we have color-coded the modules in **Figure 2C**, altered the label position and adjusted the figure description accordingly and hope this has improved the clarity of this figure (page 30, Figure 2, panels C-E).

Figure 2: Proteome response profiling of EC-cytokine interactions

[...] **C:** Profile plots of modules describing cytokine proteomic responses with cytokine annotation. Gradient scale indicated z-scores of median LFQ-score of proteins in a module per stimulus, Yellow: cytokines related to an increased abundance response profile; Purple: cytokine(s) related to a decreased protein abundance response. Cytokines which contribute to the module regulation are highlighted. Replicates (N=3-6) have been summarized to medians for visualization.[...]

7) A final remark *ME* is the nomenclature for the Module Eigenene, therefore modules should be named "*M*".

-In accordance, we have corrected all '*ME*' to '*M*'.

Reviewer #2 (Remarks to the Author):

Herein I submit a review of a manuscript entitled "Multi-omics Delineation of Cytokine-Induced Endothelial Inflammatory States" authored by Groten et al.

This study represents major effort to depict potential protein interactions involved in TNF α /INF γ -induced inflammation in endothelial cells based on the compilation of some rather impressive datasets and their interrogation. Some of these data recapitulate previous findings; however, there is sufficient novelty to warrant publication. I am very much impressed with this manuscript but I do have some concerns described below.

1. BOECs are an interesting choice of cells to conduct this study. Can the authors comment on whether these cells adopt a microvascular or macrovascular phenotype and include it in the discussion? If microvascular, these findings may be particularly significant to eye and kidney vasculopathies? Furthermore, could it be possible that some of the observed mRNA and protein changes under the stimuli described, be attributed to retention of EPC character?

We thank the reviewer for several interesting discussion points on the matter of this EC model and its outlook on potential vasculopathies and address them per question, below:

1.1. Can the authors comment on whether these cells adopt a microvascular or macrovascular phenotype and include it in the discussion?

- Unfortunately, there is no obvious answer whether BOECs are generically micro or macro vascular as they are differentiated within the in vitro microenvironment. In general their function is to repair and facilitate neovascularization at sites of the tissue damage⁵, so dependent on the microenvironment inducing their differentiation they could take on both macro- or micro-like roles. In response to reviewer 1 (see comment 2 and **Figure S5**), we have also compared our steady-state mRNA to datasets from other studies. These studies, which contained primarily macro-vascular derived EC-types did show a general overlap with the BOECs used in this study. However, there was notable variation within these EC-types as well. Therefore in this in vitro setup we deem it challenging to make a definitive statement on this point. We have added an extra comment in the discussion highlighting the unclarity of this point.

[369 – 379, page 12-13, **Discussion**] "An important distinction considering vascular inflammatory disorders, is that between the micro- and microvasculature. Although studies have shown ECs derived from the macro- or micro-vasculature retain their specific differences in culture⁷², it is difficult to assess whether BOECs take on a macro- or micro-like vessel type as they differentiate within the in vitro microenvironment. Comparing steady state mRNA data highlighted the challenges of assessing EC variation over multiple studies. We did observe overall similar relative expression levels of EC markers and TNF α and INF γ receptors, suggesting the observed cytokine responses could potentially be translated to other EC types as well³⁰⁻³². To assess whether these inflammatory states are conserved between different vascular beds is a crucial step in future studies to understand tissue-specific EC inflammation."

1.2. If microvascular, these findings may be particularly significant to eye and kidney vasculopathies?

-The suggestion that these molecular mechanisms might be of particular interest in eye and kidney vasculopathies is indeed very interesting. A study on acute anterior uveitis described an inflammatory microenvironment in which both TNF α and IFN γ were present⁶, while another study highlighted an overall increase in HLA-DR in various ocular diseases compared to healthy eyes⁷, which we also observed. As the reviewer mentions, in inflammatory kidney vasculopathies EC inflammatory markers are upregulated which could have similar underlying molecular mechanisms we describe here⁸. Therefore, we have added these studies in the introduction as examples of vascular disorders with an endothelial component.

[44 – 47, page 2, **Introduction**] “This endothelial dysfunction is implicated in several multifaceted inflammatory diseases, including Transfusion Related Acute Lung Injury (TRALI), Sepsis, Rheumatoid Arthritis, Acute Respiratory Distress Syndrome (ARDS), eye vasculopathies, Chronic Kidney Disease (CKD) and COVID-19^{10–19}.”

1.3. *could it be possible that some of the observed mRNA and protein changes under the stimuli described, be attributed to retention of EPC character?*

-The exact origin and following differentiation of BOECs is indeed not completely understood, but BOECs are thought to lose their progenitor-like character through culturing, which is seen in the expression of mature EC markers and loss of hematopoietic markers in cultured BOECs⁵. We have also previously shown that these cells maintain hallmark endothelial characteristics during culture, including cobblestone morphology, expression of endothelial-specific Weibel-Palade bodies and endothelial thrombin responses^{3,4}. Moreover, relative expression of EC markers of BOECs in this study did generally overlap with ECs from different vascular beds (**Figure S5**). Nevertheless, we cannot exclude some molecular mechanisms described here are due to our in vitro model of choice. To address this we explicitly describe the use of BOECs and its limitations in the discussion of this manuscript. [354 – 379, page 12 – 13, **Discussion**]

2. *Interestingly, ECIS experiments were conducted and the authors claimed that the BOEC-derived EC demonstrated barrier function yet only a normalized resistance was reported. What were the raw resistance readings? Typically for those EC that establish true barriers, the resistance measures are in the neighborhood of 3000 ohms. The resistance to ion flow derives from two components, one is due to cell attachment to the substratum and the other to the formation of tight junctions in the paracellular cleft. Is the relative resistance shown in figure 3b simply due to cell attachment or is a true barrier formed?? IHC to demonstrate tight -junction disorganization under the stimuli that were tested and the inclusion of Fitc-dextran permeability as an additional measure of monolayer permeability would help clarify this issue.*

-We sought to apply the ECIS as a method to highlight the functional impact of cytokine synergy. However, we appreciate the concerns raised by the reviewer. Our measured resistance fluctuated between 2000-3000 ohms, however we are now aware that it would require dedicated studies to make conclusions on barrier integrity effects of inflammatory cytokines, and that we were too quick in including these results. Because of the valid remarks of both reviewers (see also comment 5 of reviewer #1) and as it does not alter our other findings or conclusions in this study we have opted to exclude this experiment (previously Figure 3B) from the manuscript.

3. *In the methods section BOECs were cultured in 18% fetal calf serum (FCS) for the secretome analysis. I presume they switched to ECs at some point ? I also presume that the medium was changed to add the cytokine stimuli ? It would be helpful to include a few more details that may have been omitted. Along those lines, an impressive list of cytokines was detected in the conditioned medium, were these samples manipulated in some manner to deplete the 18% FCS? I checked reference 34 and it doesn't seem to suggest they were. If these samples were not depleted of this bulk protein I am surprised that you were able to detect these cytokine in the conditioned medium. Would use please clarify?*

-We apologize this was not clearly indicated in the methods. We completely agree that the use of FCS should be avoided in secretome experiments for several reasons: We cannot accurately distinguish cytokines already in the FCS to the ones excreted from the ECs and measuring samples containing FCS disrupts the MS measurements due to the high abundance of e.g. BSA. Therefore, the secretome experiments were performed in EBM medium without any additions (as were the SILAC-phospho-proteome experiments). The (phenol-red-free) medium, containing the excreted proteins from the ECs is then measured directly after concentration, digestion and desalting steps. As this was not evident in our description, we have rewritten the stimulation under separate heading in the M&M to clear up how these were performed.

[409 – 418, page 14, **Methods**] “All recombinant human cytokines used for stimulations were obtained from Peprtech (**Table S2**). ECs were stimulated with 10 ng/ml per cytokine for indicated timepoints, with exception of dose response experiments, in which cells were stimulated at 1, 10 and 100 ng/ml. Prior to stimulation cells were washed 3x with PBS and stimulations were performed in endothelial basal medium (Lonza) supplemented with 18% FCS (Bodinco) and EGM bulletkit (Lonza), with exception of SILAC BOECs and secretome experiments. SILAC BOECs were serum starved for 2 hours prior to stimulation and stimulated in endothelial basal medium (Lonza) without additions. Stimulation in secretome experiments were performed in phenol-red-free endothelial basal medium (Promocell) without any additions”.

1. Karampini, E. *et al.* Defective AP-3-dependent VAMP8 trafficking impairs Weibel-Palade body exocytosis in Hermansky-Pudlak Syndrome type 2 blood outgrowth endothelial cells. *Haematologica* **104**, 2091–2099 (2019).
2. Schillemans, M. *et al.* Weibel-Palade Body Localized Syntaxin-3 Modulates Von Willebrand Factor Secretion From Endothelial Cells. *Arterioscler. Thromb. Vasc. Biol.* **38**, 1549–1561 (2018).
3. Martin-Ramirez, J., Hofman, M., van den Biggelaar, M., Hebbel, R. P. & Voorberg, J. Establishment of outgrowth endothelial cells from peripheral blood. *Nat. Protoc.* **7**, 1709–15 (2012).
4. van den Biggelaar, M. *et al.* Quantitative phosphoproteomics unveils temporal dynamics of thrombin signaling in human endothelial cells. *Blood* **123**, e22-36 (2014).
5. Medina, R. J. *et al.* Endothelial Progenitors: A Consensus Statement on Nomenclature. *Stem Cells Transl. Med.* **6**, 1316–1320 (2017).
6. Ghiță, A. C., Ilie, L. & Ghiță, A. M. The effects of inflammation and anti-inflammatory treatment on corneal endothelium in acute anterior uveitis. *Rom. J. Ophthalmol.* **63**, 161–165 (2019).
7. Ni, M., Chan, C. C., Nussenblatt, R. B., Li, S. Z. & Mao, W. Iris inflammatory cells, fibronectin, fibrinogen, and immunoglobulin in various ocular diseases. *Arch. Ophthalmol. (Chicago, Ill. 1960)* **106**, 392–5 (1988).
8. Diaz-Ricart, M. *et al.* Endothelial Damage, Inflammation and Immunity in Chronic Kidney Disease. *Toxins (Basel)*. **12**, (2020).

REVIEWERS' COMMENTS:

Reviewer #1 (Remarks to the Author):

Dear Authors,

congrats to your nice paper !

Reviewer #2 (Remarks to the Author):

I am very pleased with authors modifications/responses to their manuscript and I have no further issues.